# Designed and validated novel allele-specific primer to differentiate Kernel Row Number (KRN) in tropical field corn

Ganapati Mukri[1]*, Kumari Shilpa[1], R. N. Gadag[1], Jayant S. Bhat[2], Chandu Singh[1], Navin C. Gupta[3], Chandra Prabha[1], Sahana Police Patil[1]

**1** ICAR-Indian Agricultural Research Institute, New Delhi, India, **2** ICAR-IARI Regional Research Centre, Dharwad, Karnataka, India, **3** ICAR-National Institute for Plant Biotechnology, New Delhi, India

* ganapati4121@gmail.com

## Abstract

### Background

Kernel row number (KRN) is an important yield component trait with a direct impact on the productivity of maize. The variability in KRN is influenced by the inflorescence meristem size, which is determined by the CLAVATA-WUSCHEL pathway. A CLAVATA receptor-like protein, encoded by the FASCIATED EAR2 (*fea2* gene), enhances the growth of inflorescence meristem and is thus involved in the determination of KRN. The amplicon sequencing-based method was employed to dissect the allelic variation of the *fea2* gene in tropical field corn.

### Methodology/Principal finding

Amplicon-based sequencing of AI 535 (Low KRN) and AI 536 (High KRN) was undertaken for the gene *fea* 2 gene that codes for KRN in maize. Upon multiple sequence alignment of both sequences, A to T transversion at the 1311 position was noticed between Low KRN and High KRN genotypes resulting in different allelic forms of a *fea2* gene in tropical maize. An allele-specific primer *1311 fea2.1* was designed and validated that can differentiate High and Low KRN genotypes.

### Conclusion/Significance

Maize has high variability for KRN and is exemplified by the wide values ranging from 8–26 KRN in the maize germpalsm. The sequence-based approach of SNP detection through the use of a specific primer facilitated the detection of variation present in the target trait. This makes it possible to capture these variations in the early generation. In the study, the PCR-based differentiation method described for the identification of desirable high KRN genotypes would augment the breeding programs for improving the productivity of field corn.

**Data Availability Statement:** All relevant data are within the paper and its Supporting Information files.

**Funding:** This study was supported by DST-SERB and there is no additional funding for the publication of the findings. All authors who contributed to the article are researchers in the Indian Agricultural Research Institute, New Delhi, India which is an organization of the Indian Council of Agricultural Research, and no monetary support shall be rendered by the parent organization to referred authors in this endeavor. "The funders had no role in study design, data collection, and analysis, decision to publish, or preparation of the manuscript".

**Competing interests:** The authors declare that they have no conflicts of interest.

## Introduction

Maize (*Zea mays* L.) is a monoecious plant that bears separate male and female inflorescences on the same plant. Understanding the genetic basis of floral architecture is drawing the attention of breeders, with the long-term goal of increasing the number of seeds per inflorescence, thereby enhancing crop yield [1]. The kernel row number (KRN) is one of the domesticated traits in maize, a floral trait that has a significant positive effect on grain yield [2]. Increasing the productivity of staple food crops is the need of the hour as food security is threatened by the growing human population and decreasing agricultural resources, including crop land and labor, through urbanization and industrialization. The current rate of increase of yield (0.9–1.3% per year) of the four major crops (rice, maize, wheat, and soybean) is insufficient to meet the food demand of the estimated nine billion people in 2050 [3]. Hence, new and impactful component traits need to be targeted in addition to grain yield *per se* to breed for higher productivity. The KRN is one such trait in field corn.

In maize, both male (tassel) and female (ear) inflorescences are produced from an axillary meristem, called inflorescence meristem (IM) which is produced from shoot apical meristem (SAM). The IMs will differentiate into spikelet pair meristems (SPM), which develops into spikelet [4]. Each SPM divides once to form two spikelet meristems, each of which gives rise to floral meristems that will produce a single kernel after pollination [5]. Hence, the variation in KRN in maize is decided by the number of axillary spikelet pair meristems (SPMs) that are formed by flanking the IM [6]. The simple hypothesis is that an increase in the size of the IM would provide additional space for SPM initiation and, thus, a higher KRN in maize. Ross et al. (2006) [7] found 10 *QTL* that explained more than 50% of the variation in the number of rows, and most of these *QTL* with additive effect. Evaluating other traits of the maize plant, Silva et al. (2004) [8] found additive genetic control for all characters, including kernel row number. Srdic et al. (2007) [9] in a diallel using Hayman's methodology found that the additive effect was more important in the expression of kernel row number. Liu et al. (2015) [10] identified 33 KRN quantitative trait loci (QTLs) representing 21 QTLs common to several population/environments. The majority of these common QTLs that displayed a large effect were additive or partially dominant. They found 70% KRN-associated genomic loci were mapped in KRN QTLs identified in this study, KRN associated SNP hotspots detected in NAM population and/ or previous identified KRN QTL hotspots. Though many genes *viz.*, *fea2*, *fea3*, *ids1*, *bif2*, *kn1*, *cg1*, *etc.*, are reported for KRN variation [11–14], these are not yet fully explored and captured in tropical maize breeding program.

The reverse genetic approach *i.e.*, targeting-induced local lesions in genomes (TILLING) was utilized and isolated weak alleles of *fea2* gene in temperate maize [14], but information on breeder-friendly marker tool was not availed for easy distinction between the alleles responsible for high and low KRN genotypes. The advent of advanced sequencing technology made possible the identification of large-scale nucleotide sequence polymorphisms and the establishment of their associations with the variation in the trait of interest [15].

The allelic variation generated due to Single Nucleotide Polymorphism (SNP)/INDELS requires a specific differentiation method to resolve the polymorphism [16]. Several approaches are available, with their own merits and demerits, to differentiate allelic variations resulting from the SNPs. The traditional method given by Newton et al. (1989) [17] to capture a single base pair mismatch is not sufficient for reliable discrimination between the two alleles with transition or transversion mutation. Drenkard et al. (2000) [18] demonstrated the modified allele-specific PCR procedure that enabled reliable detection of SNPs. In the present investigation, an effort was made to detect the SNPs linked to the KRN trait followed by the development of an allele-specific primer, which could differentiate the high and low KRN genotypes.

**Table 1. Variability parameters of KRN trait across the location and seasons.**

| Environment | Source | D.f. | MSS | Mean | Range | PCV% | GCV% | H² |
|---|---|---|---|---|---|---|---|---|
| E1 | Replication | 1 | 0.03 | 16 | 10–24 | 20.1 | 20.0 | 99.40 |
| | Genotype | 44 | 18.62** | | | | | |
| E2 | Replication | 1 | 0.34 | 16 | 10–26 | 21.6 | 21.6 | 99.60 |
| | Treatment | 44 | 22.27** | | | | | |
| E3 | Replication | 1 | 0.01 | 16 | 12–26 | 22.5 | 21.7 | 93.00 |
| | Genotype | 44 | 24.60** | | | | | |

**KRN**-Kernel Row Number, **D.F.** Degree of freedom, **MSS**-Mean Sum of Squares, **PCV**-Phenotypic Coefficient of Variation, **GCV**- Genotypic Coefficient of Variation,
**H²** Heritability in Broad-sense, * level of significance.
**#E1:***Kharif* 2018 at Indian Agricultural Research Institute, New Delhi.
**E2:***Rabi*2018-19 at Indian Agricultural Research Institute, New Delhi.
**E3:** Rabi 2018–19 at Regional Research Centre, Dharwad.

## Results

Two tropical maize inbred lines, AI 535 and AI 536 possessing contrasting KRN were sequenced for the *fea*2 gene. Both the amplicon sequences were aligned along with B73 (temperate inbred line), SNPs were detected and confirmed through translated amino acid sequences followed by its protein modeling. An effort was made to differentiate and validate identified SNP through restriction enzyme-based assay, and conventional and allele-specific primer-based approaches. The detailed results were presented in the following sections.

### Genetic variability and stability analysis of KRN trait

A total of 45 inbred lines were subjected to rigorous phenotypic analyses at two locations and three seasons for the KRN trait. The mean sum of squares due to genotypes for KRN was highly significant (Table 1).

The KRN phenotype ranged from 10–24 in E1, 10–26 in E2, and 12–26 in E3 with a mean KRN of 16 across the environments. The phenotypic coefficient of variation (PCV) was 20.1, 21.6, and 22.5 with E1, E2, and E3, respectively, while, the genotypic coefficient of variation (GCV) was 20.0, 21.6, and 21.7 with E1, E2, and E3, respectively. The heritability (broad-sense) was 0.99, 0.99, and 0.93 percent in E1, E2, and E3, respectively. AMMI analysis carried out in our lab (S1 Table) showed that out of 45 inbred lines, AI 535 with the lowest KRN (12) and AI 536 with a high KRN (18), showed stable expression across the tested environments (Table 2 and Fig 1). Similar, observations were also noticed by Chetan et al., 2021[19].

### Identification of SNPs in fea2 gene of high and low KRN genotype

The exploration of the sequencing technology enables the identification of variations linked to the trait at the single-nucleotide level. Since the inbred lines used in the present investigation

**Table 2. Mean and range for KRN trait of selected genotypes across environments.**

| S.No | Environment/ Genotypes | E1 | | E2 | | E3 | | Pooled |
|---|---|---|---|---|---|---|---|---|
| | | Mean | Range | Mean | Range | Mean | Range | Grand mean |
| 1 | AI 535 | 10 | 10–12 | 10 | 10–12 | 12 | 12–14 | 12 |
| 2 | AI 536 | 18 | 18–20 | 16 | 16–18 | 18 | 18–20 | 18 |

# E1, E2 and E3 (abbreviation) are similar to Table 1.

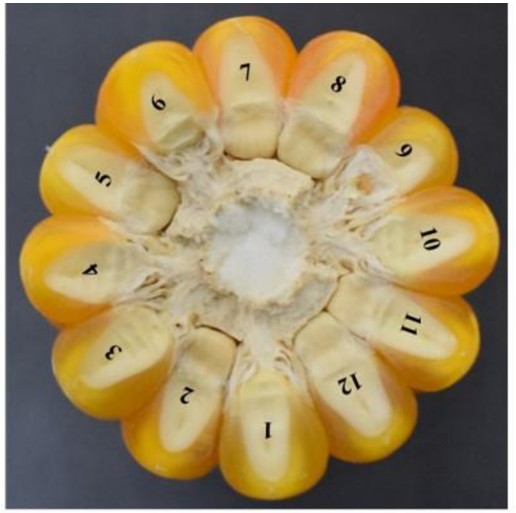 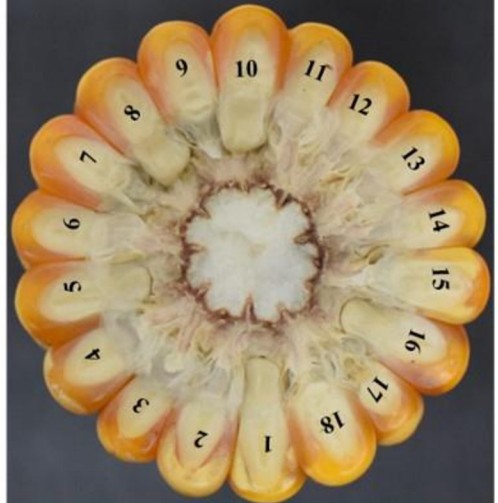

Low Kernel Row Number (KRN) line (AI 535) High Kernel Row Number (KRN) line (AI 536)

**Fig 1. Contrasting parents for high and low KRN trait.**

are of tropical origin, it is hypothesized that the KRN gene, *fea2*, does have allelic variation. The *fea2* is 2489 bp in length and has one intron and two exons that code for 593 amino acids. The *fea2* also has extra-cytoplasmic LRRs, a transmembrane domain, and a short cytoplasmic tail, which acts as a receptor at the plasma membrane [20]. To understand the allelic variation, the coding sequence (CDS) of *fea2* was targeted to capture the SNPs among AI 535 and AI 536. The PCR amplicon of the 1842 bp (S1 Fig) obtained from the CDS region that includes 5′ UTR region (56 bp) was sequenced in both AI 535 and AI 536 lines. The sequence alignment results with the *fea2* reference sequence of B73 have shown a total of eight SNPs in exonic regions. Out of eight SNPs, five SNPs were found to be neutral to amino acid changes, but the rest of the three SNPs appear to be causing non-synonymous amino acid variation between high and low KRN genotypes, such as G>747>A (Glycine-Serine), A>1311>T (Serine-Cysteine) and G>1386>C (Valine-Leucine) (Table 3 and Fig 2). Further, to understand the effect of these changes on protein structure, homology modeling was undertaken.

**Table 3. Identification of SNPs in *fea*2 gene between high and low KRN genotypes.**

| S.No | Position | Reference | Alternate | Genotype | | Reference Nucleotide | Alternate Nucleotide | Reference Amino acid | Alternate Amino acid |
|---|---|---|---|---|---|---|---|---|---|
| | | | | AI 535 | AI 536 | | | | |
| 1 | 233 | G | T | G | T | GCG | GCT | Alanine | Alanine |
| 2 | 329 | T | C | C | C | GTT | GTC | Valine | Valine |
| 3 | 611 | G | A | A | G | GCG | GCA | Alanine | Alanine |
| **4** | **747** | **G** | **A** | **G** | **A** | **GGT** | **AGT** | **Glycine** | **Serine** |
| 5 | 1139 | A | G | G | G | TTA | TTG | Leucine | Leucine |
| **6** | **1311** | **A** | **T** | **T** | **A** | **TGT** | **AGT** | **Cysteine** | **Serine** |
| **7** | **1386** | **G** | **C** | **C** | **C** | **GTT** | **CTT** | **Valine** | **Leucine** |
| 8 | 1466 | G | A | A | A | GGG | GGA | Glycine | Glycine |

Five SNPs (S. No. 1,2,3,5 & 8) were found to be neutral amino acid changes between the High and Low KRN genotypes.

Three SNPs (S.No. 4, 6 & 7) appear to be causing non-synonymous amino acid variation between the High and Low KRN genotypes.

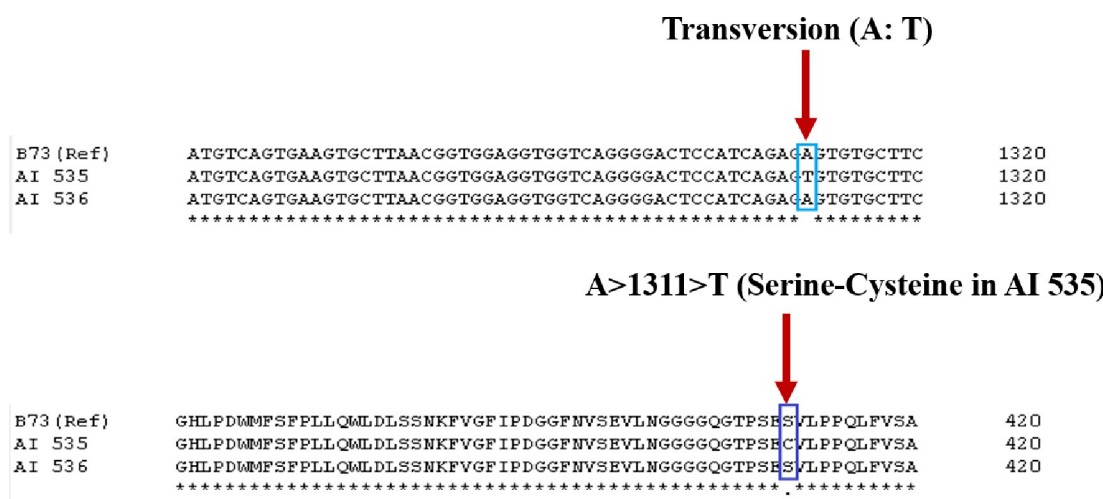

**B73 (Ref):** Reference sequence of B73, **AI 535:** Inbred lines with low KRN (12), **AI 536:** Inbred line with high KRN (18), **1320:** Is the position of nucleotide sequence of *fea2* gene

**Fig 2.** (a) Comparative nucleotide sequence of test and reference genotypes showing transversion (A: T) at 1311bp of *fea2* gene, (b) Comparative amino acid sequence of test and reference genotypes showing conversion of cysteine/serine due to A: T transversion.

## Homology modeling of protein structure and restriction enzyme-based assay

The open reading frames (ORF) identified through FGENESH in both low and high KRN genotypes were subjected to the SWISS Model. It was observed that the model LRR receptor-like serine/threonine-protein kinase FLS2 (4mna.1. A) had higher percent similarity (Fig 3) with the proposed model which may be triggered by the mutation A>1311>T (Fig 2), that changes Serine to Cysteine (Fig 2). The remaining non-synonymous SNPs did not show remarkable changes in the functionally equivalent activity of the protein or enzymes coded by them [21–24].

To capture this variation, the nucleotide sequences at nearby 1311 nucleotide positions were subjected to the identification of enzyme-based polymorphic restriction sites. A total of 13 restriction sites (S2 Table) were observed in the 400 bp (1150–1550 bp) region of the *fea2* gene in both AI 535 and AI 536. However, the identified sites were monomorphic in nature (S2 Fig). Hence, these restriction sites could not be used to differentiate high and low KRN genotypes. Therefore, PCR based differentiation assay was carried out.

## KRN genotypes and conventional primers

For differentiating the allelic variation due to A to T transversion at the 1311 position in the *fea2* gene, the eight different conventional primers designed were found non-informative among the genotypes. Though these primers showed different amplicons within each genotype, however, it amplified a similar amplicon among the test genotypes (S3 Fig). Hence, it was concluded that designed conventional primers were unable to differentiate the high and low KRN genotypes with a single nucleotide difference among them. Further, the principle of introduction of mismatch nucleotide in the forward primer to make a set of primers to locate specific changes was followed.

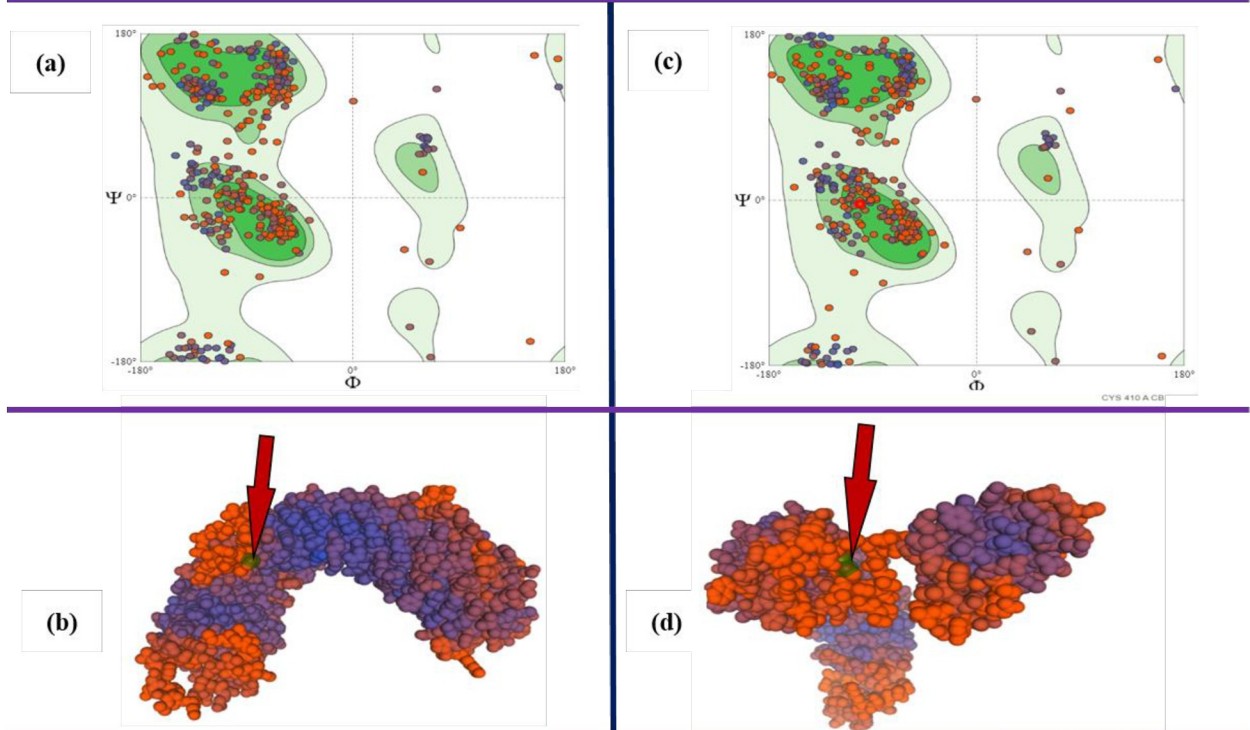

*Fig. (a) & (b)* Ramachandran plot and proposed protein model of *fea2* gene with a sequence identity of **29.94%** with template LRR receptor-like serine/threonine-protein kinase for Low KRN (AI 535), *Fig. (c) & (d)* Ramachandran plot and proposed protein model of *fea2* gene with a sequence identity of **30.53%** with template LRR receptor-like serine/threonine-protein kinase for Low KRN (AI 536)

**Fig 3. Ramachandran plot and proposed model for *fea2* gene for high (AI 536) and Low (AI 535) KRN trait.** *(a) & (b)* Ramchandran plot and proposed protein model of *feq2* gene with a sequence identity fo **29.94%** with template LRR receptor-like serine/threonine-protein kinase for Low KRN (AI 535), *(c) & (d)* Ramchandran plot and proposed protein model of *fea2* gene with a sequence identity of **30.53%** with template LRR receptor-like serine/threonine-protein kinase for Low KRN (AI 536).

## Differentiation of high and low KRN genotype through allele-specific primer

In the present study, the challenge was to differentiate high and low KRN genotypes based on the single-nucleotide difference (A to T transversion) present at the 1311 nucleotide position of the tested genotype. The primers with the 3$^{rd}$ base as G at 3′ end instead of A in the AI 536 sequence were designed to amplify the targeted nucleotide variation at the 1311 position. The AI 535 which has T instead of A at the 1311 position did not show the PCR amplification with the site-specific primers whereas, AI 536 showed amplification of the 694 bp allele (Fig 4). Hence, with the current finding we propose the primer set: 5′TGGTCAGGGGACTCCATCA G*G*G**A**3′ and 3′TGCAGACCAGAGTCGCTCGAAC5′ could be the candidate primer to differentiate high and low KRN genotypes based on the *fea2* allelic variation which is henceforth named as *fea2.1* and the primer specific to this is designated as 1311*fea2.1*. The same primer set was also used to differentiate the other genotypes having low KRN (AI 505) and high KRN (AI 542), involved in the active maize breeding program of the IARI. It was observed that primer 1311*fea*2.1 is polymorphic among AI 505 and AI 542 in relation to the 694bp allele (Fig 4). Hence the F$_2$ progenies of these two contrasting parents were utilized for primer validation.

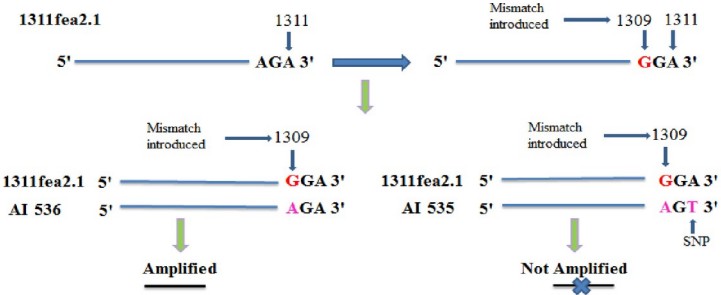

(a) Introduce the mismatch

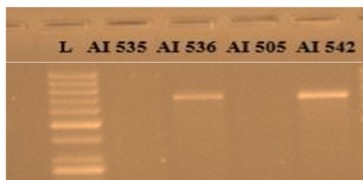

(b) L-Ladder: 50bp, AI-535 low KRN inbreds (Not amplified), AI 536 & AI 542: High KRN inbred amplified

**Fig 4. Allele-specific amplification of 1311*fea*2.1 in high and low KRN genotypes (AI 535 and AI 505: Low KRN, AI 536 and AI 542: High KRN, L: 50bp ladder).** (a) Introduce the mismatch. (b) L-Ladder: 50bp, AI-535 low KRN inbreds (Not amplified), AI 536 & AI 542: High KRN inbred amplified.

## Validation of allele-specific designed primer (1311*fea*2.1) in the F$_2$ population

The 155 individuals of the F$_2$ population derived from the crosses between AI 505×AI 542 were used for the validation of the designed allele-specific primer, 1311fea2.1. It was found that out of 155 individuals, a total of 43 individuals having high KRN phenotype showed expected amplification of 694 bp allele, and the rest of the individuals with low KRN did not show any amplification (Fig 5). Both phenotypes (high and low KRN) and genotype (presence and absence of 694 bp allele) ratio with respect to KRN is following Mendelian segregation ratio of 1:3 and hence, the designated primer, 1311fea2.1, is linked to the trait of interest.

## Discussion

Genetic improvement of KRN, being an important yield component trait that directly influences the productivity of maize, has been included as one of the major objectives in the maize breeding programs. In cultivated maize, a wide range of KRN (8–26) is available, which forms a potential genetic source for kernel trait-specific breeding in maize [25]. The KRN trait has high heritability among the yield component traits and its expression takes place in a discrete manner, hence, it is designated as a threshold trait [26]. To breed for higher KRN, phenotypic differentiation followed by selection for KRN may be cumbersome and resource-demanding. Allelic variation present within the gene of interest for the targeted trait allows selection of effective variants of a given trait. Among the cloned gene responsible for KRN, *fea*2 plays a major role in inducing KRN variation in maize. Understanding and capturing the allelic variation for KRN in temperate vis-a-vis tropical maize gives molecular insights into the trait. However, a breeder-friendly marker system is required that makes it easier the differentiation of the variants for their utilization in plant breeding programs.

In maize, many genes *viz.*, *fea*2, *fea*3, *fea*4, *etc.*, have been reported to underlay the expression of KRN traits. Bommart et al. (2013) [14], isolated a weak allele of *fea*2-1328 (1430 C>T)

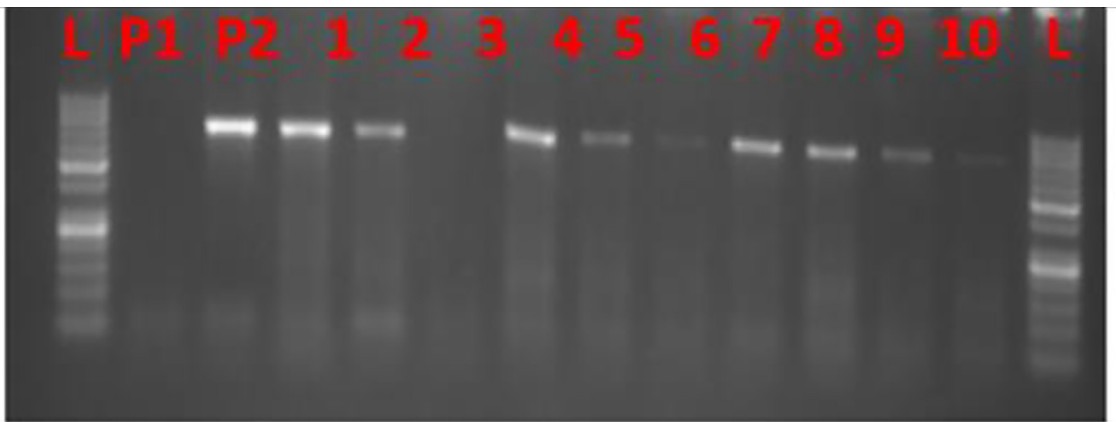

L: 50 bp ladder, P1: AI 505, P2: AI 542,
1,2,4,5,6,7,8,9,10: High KRN, 3: Low KRN

**Fig 5. Validation of novel primer 1311fea2.1 in 155 F$_2$ progenies.**

through TILLING (targeting-induced local lesions in genomes) approach, and they proposed that this allele can increase inflorescence meristem size and KRN without reducing ear length or triggering fasciation, but both of which would be detrimental to yield. A similar ear fasciation phenotype with a high KRN genotype (AI 536) is available at ICAR-IARI, New Delhi. To understand the allele present and to capture/differentiate the same for future usage, a amplicon sequence-based approach was followed, which showed that nucleotide position at 1311bears an A-T transversion (Fig 2) that is expected to produce amino acid change (Serine to Cysteine) in low (AI 535) *vis-à-vis* high KRN genotypes (AI 536) (Fig 2). The amino acid change was confirmed by subjecting the sequence to protein modeling. Serine is a necessary component of proteins and a precursor to various biomolecules, including other amino acids such as trypto-phan [27]. Serine also acts as a polar transporter of auxin that phosphorylates BARREN STLK (BA 1) which is encoded by *barren inflorescence 2* (*bif2*). The serine/threonine-protein kinase is involved in axillary meristem initiation, which finally decides the number of kernel rows in maize [4,25]. It was proposed earlier also that serine/threonine-protein kinase is responsible for variation in kernel traits in maize [28]. Hence, nucleotide position A>1311>T was a candidate to differentiate between high (AI 536) and low (AI 535) KRN genotypes.

The availability of restriction sites among the sequences allows tagging the allelic variation [29]. But in the present situation, the region around the nucleotide sequence 1311 of *fea2* (440bp) in AI 535 and AI 536 did not have any polymorphic single restriction site, due to the presence of identified SNP (A/T) (S2 Fig), which makes it difficult to differentiate both the genotypes. Hence, other methods to differentiate the allelic variation were explored.

According to the traditional SNP genotyping method, single nucleotide changes can be detected using the specific PCR primers designed such that the 3′terminal nucleotide of a primer corresponds to the site of the SNP [30]. It was expected that 3′ termini are extended by *Taq* polymerase with much lower efficiency than correctly matched termini, hence preferential amplification of specific alleles [31]. Similarly, Newton *et al.*, 1989 proposed a method to design random primers using flanking sequences around the target position of the gene [17].

In the study, a similar method was followed to design a set of 8 primers for targeting specific amplification of the 1311 nucleotide position of *fea2*. However, these sets of primers could not

differentiate contrasting parents through amplification, implying that a single base mismatch at the specific allele is usually not reliable for discrimination between two alleles. Such results were reported earlier as well [16]. To overcome this problem, a modified method to design allele-specific primers having artificially introduced mismatch to increase the specificity and reliability of discrimination between alleles has been suggested [16,32]. Different mismatched principles were proposed by different investigators to improve the allele specificity of the primers [18]. For instance, it was observed that SNPs (A/T) containing CA mismatches in the 3rd nucleotide from the 3′ end of the primers had the highest allele specificity [31]. But, Hirotsu et al. (2010) [33], proposed that A-T transversion and A-G transition were useful base pair mismatches for the improvement of allele-specific amplification. In the present study, the genotype with high KRN had the last three nucleotides as A, G, A, while the low KRN genotype had A, G, T at 1309, 1310, and 1311 positions, respectively. Considering the mismatch principles, nucleotide 'G' was introduced in the forward primer as 5′TGGTCAGGGGACTCCATCAG**G**GA3′ corresponding to 1309 position of high KRN (AI 536)/reference genotype (B73). The non-specific reverse primer was designed as 3′TGCAGACCAGAGTCGCTCGAAC5′ taken from the other side of the SNP [34]. Upon amplification, AI 536 showed the presence of a 694bp allele in contrast to AI 535 where it was absent (Fig 4). The induced mismatch unaltered the affinity of primer ligation at the 1311 position in AI 536 compared to others, as AI 535 which has T instead of A in nucleotide position 1311 offers two mismatches for the primer to ligate, hence there is no amplification of targeted allele. Therefore, the primer set: 5′TGGTCAGGGGA CTCCATCAG***G***G**A**3 and 3′TGCAGACCAGAGTCGCTCGAAC5′ can be a candidate primer to differentiate 1311nucleotide position in *fea*2 gene and thereby high and low KRN genotypes. It was designated as an allele-specific primer and named as 1311*fea*2.1.

Further, the primer 1311*fea*2.1 was used on AI 505 and AI 542 which were low and high KRN genotypes, respectively. The genotype AI 505 did not show amplification of the 694 bp allele; it indicated the presence of SNP 'T' instead of 'A' at the 1311 nucleotide position. The $F_2$progenies from the cross (AI 505 × AI 542) were subjected to phenotyping for KRN and it was observed that out of 155 $F_2$ progenies, 43 were high KRN, and 112 were low KRN. This $F_2$ population was used to validate the results of the 1311*fea*2.1 primer. Interestingly, all the 43 individuals categorized as high KRN showed the amplification 694 bp allele with SNP 'A' and the remaining individuals with low KRN did not show the amplification implying the absence of the specific allele (Fig 5). In addition, the total of 197 diverse germplasms was characterized for *1311fa2.1* primer (Fig 6).

The correlation between the presence of marker and KRN phenotype indicated a high and significant correlation (r = 0.87) among them. The regression analysis also indicated a significant association between KRN and primer ($R^2$ = 0.75). This clearly confirmed the strength of the 1311*fea*2.1 primer in differentiating high and low KRN genotypes in the studied population. Since the designed specific primers showed simple Mendelian inheritance and effectively differentiated the high KRN genotype with low and high, this can be an effective candidate marker to select high KRN genotypes at the early segregating generation of maize having a varied kernel row number followed by effective generation advancement of desired KRN trait and thereby reducing the breeding cycle. The gene *fea2* showed a different allelic form as was evident from the earlier study also [27]. The novel allelic variation identified in the study also may be specific to tropical germplasm, controlling the KRN trait.

## Materials and methods

**Field experiment design and implementations.** A set of 45 tropical maize inbred lines (S3 Table) were evaluated in a randomized complete block design with two replications across

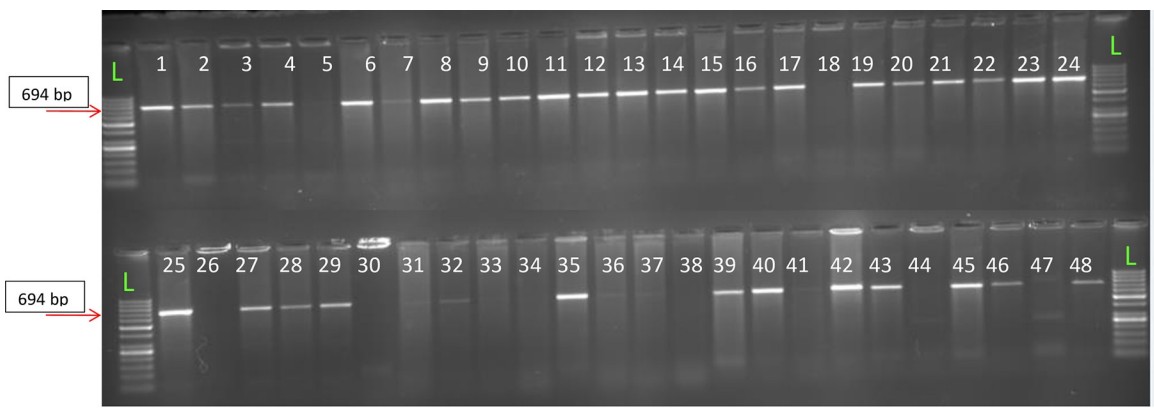

Genotypes: 1=C-18, 2=C-84, 3=D-2282-1, 4=CDM-550, 5=BGD-4814, 6=PDM-24-4, 7=CDM-554, 8=KRN-114 , 9=C-11, 10=21100393, 11=BM-1472, 12=C-83, 13=C-139,   14=PML-68(W),  15=C-2809-1-2, 16=DIM-316, 17=21100389, 18=LM-14, 19=21100404, 20=21100388, 21=DD2313-B-1, 22=PDM-4251, 23=21100390, 24=C-10, 25=C-12, 26=PDM-10, 27=D-2332-4, 28=DDM-313, 29=C-2752, 30=BLSB-7, 31=21100385, 32=D-2386-B-2, 33=CDM-320-1, 34=21100403, 35=D-2569-A, 36=PML-18-3, 37=PDM-6555, 38=21100387, 39=AI-564, 40=AI-575, 41=PDM-4591 (W), 42=AI-574, 43=AI-569, 44=AI-580, 45=AI-582, 46=AI-566, 47=AI-577, 48=AI-556. L=50 bp

**Fig 6. Validation of novel primer 1311*fea*2.1 in germplasm varying for KRN.**

the three environments (E1, E2, and E3), i.e. one location during the rainy season of 2018 at ICAR-Indian Agricultural Research Institute (ICAR IARI), New Delhi (E1) and during the post-rainy season of 2018–19 at two locations, viz., ICAR-IARI, New Delhi (E2) and ICAR-IARI, Regional Research Centre, Dharwad (E3). Each genotype was sown in two rows of three meters in length with a spacing of 75×20 cm. The recommended agronomical package of practices was followed to raise a healthy crop. The data obtained on KRN in each tested location and season were subjected to statistical analysis using SAS 9.3 v (SSCNARS, IASRI, New Delhi). Out of 45 inbred analyzed, two genetically fixed lines *viz.*, AI 535 and AI 536, possessing contrasting phenotypes for KRN trait, with 12 and 18 KRN, respectively, were subjected to further investigation.

**Genomic DNA isolation.**   The genomic DNA of the inbred lines, AI 535 and AI 536, was isolated using CetylTrimethyl Ammonium Bromide (CTAB) method [35] with suitable modification.

## Amplicon sequencing of *fea*2 gene

The PCR primers were designed from the reference sequence of the gene *fea2* available in Maize-GDB (https://www.maizegdb.org/) with gene ID: GRMZM2G104925 from 5′ untranslated region (5′ UTR) by using DNA Club primer designing software (http://www.softwaresea.com/Windows/install-DNA-Club-10449725.htm)with the standard parameter for primers (primer length 20–24 bp, GC content 40–60% and tm 56˚C-60˚C). The PCR reaction was set up with a customized thermal cycler program of primary denaturation at 95˚C for 4 minutes followed by 35 cycles of denaturation at 94˚C for 30 sec, annealing at 65˚C for 50 sec/kb, extension at 72˚C for 1 min with final extension for 10 min at 72˚C. The PCR-amplified products were resolved on a 4% agarose gel (HiMedia). The amplicons were sequenced using the Illumina sequencing 2500 platform (AgriGenome Labs Pvt Ltd,Cochin, Kerala, India). The obtained sequences were pre-processed with a customized bioinformatics pipeline.

The read alignment was performed using Burrows-Wheeler Aligner (BWA-Ver. 0.7.12) program with default parameters. Then, read sorting was done using the Picard tool with Sort-Sam command. Duplicate sequences were removed using the Picard Mark Duplicates

command. The generated gene sequences were aligned through ClustalOmega (https://www.ebi.ac.uk/Tools/msa/clustalo/) for multiple sequence alignment along with the B73 reference sequence. The variant-calling was performed to identify SNPs and/or INDELs. These SNPs and/or INDELs were visualized through SNAPGENE viewer with default parameters (https://www.snapgene.com/snapgene-viewer/). The Gene Structure Display Server (GSDS 2.0) was used to identify the exonic as well as the intronic regions of the genes (http://gsds.gao-lab.org). The SNPs and/or INDELs present only in the exonic regions were considered for the analysis of transitional change using FGENESH software. The amino acid sequences present in the open reading frame were aligned using ClustalOmega and identified the variation in amino acid sequences.

## Homology modeling of protein structure

Gene Sequences of Open Reading Frame (ORF) with SNPs were used for protein homology modeling through SWISS-MODEL (https://swissmodel.expasy.org). The coding nucleotide sequences were pasted in the interface (FASTA format), which generated a Ramachandran plot and protein 2D/3D model.

## Identification of restriction sites

The NEB cutter (http://tools.neb.com/NEBcutter), a web-based program, was used to find out the restriction site/s at the specific position, where allelic variation is present in the nucleotide query sequences and also find the other enzymes with single restriction sites in the given sequence (S2 Table). The 400 bp nucleotide sequences of *fea*2 harbored SNP both in AI 535 and AI 536 were analyzed to locate large, non-overlapping open reading frames using the *E. coli* genetic code.

**Conventional primers to differentiate the SNPs at the 1311 nucleotide position.** There was as an A to T transversion at the 1311 nucleotide position of tested genotypes which probably differentiates high KRN genotypes from low KRN. Primers were designed using DNAClub software to differentiate allelic variation, in such a way that either the 3′ ends of the forward primer should harbor the SNPs or primers should amplify the region inclusive of the 1311 nucleotide position of the *fea2* gene. A set of eight primers (S4 Table) were designed and preliminary screening was done to search sequence homology through National Center for Biotechnology Information (NCBI) database using the BLASTN tool (https://blast.ncbi.nlm.nih.gov/Blast.cgi?PAGE_TYPE=BlastSearch). PCR was performed with all eight primer pairs in the contrasting parents of KRN trait with initial denaturation 95˚C for 4 minutes followed by 35 cycles of denaturation at 95˚C for 30 sec, annealing at60˚C for 1min with final extension for 5 min at 72˚C using 35 cycles) using OnePCRTaq polymerase (GeneDireX).

## Specific primer to differentiate the SNPs at 1311 nucleotide position

To differentiate the 1311 nucleotide position (A-T transversion), a forward primer (5′ TGGTCAGGGGACTCCATCAG**G**GA 3′) was designed using DNAClub software, by artificially introducing a mismatch **G** at 3rd bp from 3′ end of the SNP, which causes a non-synonymous change in amino acid sequence. Standardized PCR amplification (initial denaturation 95˚C for 4 minutes followed by 35 cycles of denaturation at 95˚C for 30 sec, annealing at 66˚C for 45 seconds with final extension for 5 min. at 72˚C) was done using reverse primer 3′ TGCAGAC CAGAGTCGCTCGAAC 5′.

## Validation of the 1311*fea*2-1 SNP-specific primer in a bi-parental population

The $F_1$ hybrid (AH-4500) was generated using contrasting parents for KRN, AI 505 (12 KRN), and AI 542 (22 KRN) during the rainy season, 2019, and was advanced to $F_2$ during post rainy season, 2019–20 and a total of 155 $F_2$ plants were raised using a recommended package of practices during the rainy season, 2020 at ICAR-IARI, New Delhi. Each $F_2$ plant was tagged and genomic DNA was isolated as per the CTAB method [35]. The primer *1311fea2.1* was employed for PCR-based amplification in parental inbred lines as well as in 155 $F_2$ plants (PCR procedure was followed as described in the section: Specific primer to differentiate the SNPs) up on their phenotypic characterization for KRN (S5 Table). The amplification profile was recorded as the presence or absence of a 694 bp amplicon. A total of 197 diverse inbred lines were profiled with the primer *1311fea2.1*, up on phenotypic characterization of their KRN (S6 Table). The correlation between the presence of marker and variation in the KRN trait was significant (r = 0.866). The association of primer and trait was also highly significant ($r^2$ = 0.751).

## Supporting information

**S1 Fig. A Gel picture is representative of complete amplification of target *fea*2 gene.**
(TIF)

**S2 Fig.** Monomorphic restriction site of both AI 535 (a) and AI 536 (b) around the detected SNPs (A/T) at 1311 nucleotide position. (a) Monomorphic restriction site of AI 535. (b) monomorphic restriction site of AI 536.
(TIF)

**S3 Fig.** Position of conventional primers at around SNP (A/T) and their monomorphic amplification in A: AI 535 and B: AI 536 inbred lines.
(TIF)

**S1 Table. Mean yield, IPCA1, IPCA2, ASVi and ASVi rank for genotypes over three locations.**
(DOCX)

**S2 Table. Monomorphic restriction site present in both AI 535 and AI 536 at 440 bp (1161) around the 1311nucleotide position (inclusive) in the *fea*2 gene.**
(DOCX)

**S3 Table. Pedigree of inbred lines used in this study.**
(DOCX)

**S4 Table. List of nonspecific primer(s) designed to target 1311nucleotide position of *fea2* gene.**
(DOCX)

**S5 Table. Amplification status of 1311*fea*2.1 in $F_2$ population (AH-4500 $F_2$).**
(DOCX)

**S6 Table. Amplification status of 1311*fea*2.1 in 197 germplasm.**
(DOCX)

## Acknowledgments

The author (s) is/are thankful to the Director, ICAR-Indian Agricultural Research Institute (ICAR-IARI), New Delhi, India, and Department of Science and Technology-The Science and

Engineering Research Board (DST-SERB), Govt. of India, New Delhi, India, for the financial support extended to conduct the experiment.

## Author Contributions

**Conceptualization:** Ganapati Mukri.

**Data curation:** Ganapati Mukri, Chandu Singh.

**Formal analysis:** Jayant S. Bhat, Chandra Prabha.

**Methodology:** Ganapati Mukri, Navin C. Gupta.

**Software:** Navin C. Gupta.

**Supervision:** Ganapati Mukri, Jayant S. Bhat.

**Validation:** Kumari Shilpa, Sahana Police Patil.

**Visualization:** Chandu Singh, Navin C. Gupta, Chandra Prabha.

**Writing – original draft:** Ganapati Mukri.

**Writing – review & editing:** R. N. Gadag, Jayant S. Bhat, Chandu Singh.

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
