## [Decision Letter · Decision Letter 0]

29 Jul 2022

PONE-D-22-16054Designed and Validated Novel Allele-Specific Primer to Differentiate Kernel Row Number (KRN) in Tropical Field CornPLOS ONE

Dear Dr. Mukri,

Thank you for submitting your manuscript to PLOS ONE. After careful consideration, we feel that it has merit but does not fully meet PLOS ONE’s publication criteria as it currently stands. Therefore, we invite you to submit a revised version of the manuscript that addresses the points raised during the review process.

Based on the reviewers' comments and my own assessment, the manuscript need a major revision in all sections of the manuscript due either English language, grammatical errors or lack of relevant information and discussion. There is not much background information regarding previously reported QTLs/genes for KRN in corn. Since the panel is genetically diverse, any pedigree information of the 45 inbred lines used in this study should be provided as a supplementary table. The tables 1 and 2 were mentioned in results section but could not be found. Since number of genotypes used is too low for validation, it is advisable to use larger and more diverse population. The authors should discuss how the allelic variation in fea2 gene is controlling KRN trait in corn. The manuscript pages and lines should be numbered  in the revision. Please submit your revised manuscript by Sep 12 2022 11:59PM. If you will need more time than this to complete your revisions, please reply to this message or contact the journal office at plosone@plos.org. Please include the following items when submitting your revised manuscript:A rebuttal letter that responds to each point raised by the academic editor and reviewer(s). You should upload this letter as a separate file labeled 'Response to Reviewers'.A marked-up copy of your manuscript that highlights changes made to the original version. You should upload this as a separate file labeled 'Revised Manuscript with Track Changes'.An unmarked version of your revised paper without tracked changes. You should upload this as a separate file labeled 'Manuscript'.

We look forward to receiving your revised manuscript.

Kind regards,

Prasanta K. Subudhi, Ph.D.

Academic Editor

PLOS ONE

Journal Requirements:

"SERB: EEQ/2016/00047"

"No authors have competing interest"

Additional Editor Comments:

Major revision

Reviewers' comments:

Reviewer's Responses to Questions

**Comments to the Author**

1. Is the manuscript technically sound, and do the data support the conclusions?

Reviewer #1: Yes

Reviewer #2: Partly

2. Has the statistical analysis been performed appropriately and rigorously? 

Reviewer #1: Yes

Reviewer #2: Yes

3. Have the authors made all data underlying the findings in their manuscript fully available?

Reviewer #1: Yes

Reviewer #2: Yes

4. Is the manuscript presented in an intelligible fashion and written in standard English?

Reviewer #1: Yes

Reviewer #2: No

5. Review Comments to the Author

Reviewer #1: Manuscript by Mukri et represents an interesting study identifying the novel allele-specific primer 1311 fea2.1 of fea2 gene regulating the Kernel row number (KRN) in maize. The research work involved the identification of SNPs and its validation via genomics and genetics approach. The title clearly reflects the content; the abstract is sufficiently; informative appropriate keywords are given. However, I do have some concerns, primarily to do with results and discussion

1. How much reproducible the SNPs are? Does it corresponds to other high KRN genotypes?

2. It is not clearly defined (or at least any hypothesis) how allelic variation of the fea2 gene controls the KRN trait? For example, effect on CLAVATA-WUSCHEL pathway gene expression!!

Reviewer #2: 1- Introduction need to re-write: Most of the sentences are too long and difficult for reader to understanding well. So, I suggest to re-write some paragraphs with shorter sentences.

2- In Introduction, there is no information about the previously reported gene location, QTLs/Genes…etc about the KRN in corn.

3- Is there any information (pedigree/ source) of the 45 maize inbred lines that used for field evaluation? This is very important to see how much this panel is genetically diverse. Please present it as supplementary Table.

4- In the results the section “Genetic variability and stability analysis of KRN trait”

Where are the Table 1 Table 2? I couldn’t find them in text.

5- Overall, 45 genotypes are too small population for validating the specific-primer for ant traits. So, I recommend to check this specific-primer on bigger and diverse population or in an segregating population.

6. PLOS authors have the option to publish the peer review history of their article (what does this mean?). If published, this will include your full peer review and any attached files.

Reviewer #1: No

Reviewer #2: No

---

## [Author Response · Author response to Decision Letter 0]

30 Sep 2022

As per the suggestions, the required modifications were done. We have provided in supplementary table 3. Query about table 1 and 2 was attempted. We validated 197 genotypes for the association of primer with the trait. correlation and regression analysis was done and presented in table 4.

we have discussed importance of allelic variation and information on QTLs as suggested.

---

## [Decision Letter · Decision Letter 1]

18 Oct 2022

PONE-D-22-16054R1Designed and Validated Novel Allele-Specific Primer to Differentiate Kernel Row Number (KRN) in Tropical Field CornPLOS ONE

Dear Dr. Mukri,

Thank you for submitting your manuscript to PLOS ONE. After careful consideration, we feel that it has merit but does not fully meet PLOS ONE’s publication criteria as it currently stands. Therefore, we invite you to submit a revised version of the manuscript that addresses the points raised during the review process.

 In view of the comments and issues of the reviewer 3, I am asking you to address those in your revised manuscript. I agree with the concerns raised by the reviewer regarding lack of details and the English language issues. The allele specific amplification concerns should be addressed because that is the only important section of the manuscript. 

We look forward to receiving your revised manuscript.

Kind regards,

Prasanta K. Subudhi, Ph.D.

Academic Editor

PLOS ONE

Additional Editor Comments (if provided):

Major revision

Reviewers' comments:

Reviewer's Responses to Questions

**Comments to the Author**

1. If the authors have adequately addressed your comments raised in a previous round of review and you feel that this manuscript is now acceptable for publication, you may indicate that here to bypass the “Comments to the Author” section, enter your conflict of interest statement in the “Confidential to Editor” section, and submit your "Accept" recommendation.

Reviewer #2: All comments have been addressed

Reviewer #3: (No Response)

2. Is the manuscript technically sound, and do the data support the conclusions?

Reviewer #2: Yes

Reviewer #3: Partly

3. Has the statistical analysis been performed appropriately and rigorously? 

Reviewer #2: Yes

Reviewer #3: No

4. Have the authors made all data underlying the findings in their manuscript fully available?

Reviewer #2: Yes

Reviewer #3: No

5. Is the manuscript presented in an intelligible fashion and written in standard English?

Reviewer #2: Yes

Reviewer #3: No

6. Review Comments to the Author

Reviewer #2: (No Response)

Reviewer #3: In this manuscript, the authors simply selected a candidate gene fea2.1 and sequenced the gene from two genotypes with variation in kernel row number. Then, primers were designed to validate in a segregating population

The authors mentioned next generation sequencing method several times in this manuscript (page 12, 14). You did not use next generation sequence based approach as mentioned in page 14.In real case, they just used Sanger’s sequencing method to detect the difference between two alleles. This should be changed in their revised manuscript.

Presence and absence is not the surest way to validate the association between SNP and trait. In this manuscript, they could identify the 694 bp alleles in AI 536 but no amplification in AI 535. This no amplification could be due to human error or PCR artifact. Therefore, it is better to amplify the AI 535 allele so that both homozygotes and heterozygotes can be amplified. Since this is not a difficult to do, I am suggesting the authors to do this experiment for acceptance of this manuscript. You may have to design another set of primers for amplifying AI 535 allele.

Why did you use AI 535 (lowest KRN, 12) and AI 536 (highest KRN, 18)? Is this the most contrasting pair?

The manuscript lacks details in the materials and methods.

For example: The name of 197 diverse inbred lines and their phenotypic data should be provided. Similarly, there is no information provided for the F2 population used for validation.

Fig. 2, 4, and 6 should go to Supplementary information. If the restriction site is identical, there is no point keeping as a Figure in the Main Text.

Table 4 should be deleted. There is no need of giving a single correlation coefficient as a table. It can be described in the text.

There are language issues which need to be corrected.

Introduction: The current rate…….are insufficient: Here it should be a singular verb ‘is’

Page 12: use one decimal for PCV and GCV (tables and Text); Heritability should be revised as 0.99, 0.99, and 0.93 (in table and text).

Please put space between two words: example at1311 should be at 1311; 542were should be 542 were. There are also other places which should be checked. 1saghaimaroof55?, 2019-20and etc.

Page: 14: ‘he proposed’ should be ‘they proposed’ since there are multiple authors.

Page 16: ‘mismatch principles’ should be ‘mismatched principle’.

Page 18-19; subheading ‘Conventional primers to locate the SNPs at 1311 nucleotide position’ and ‘Specific primer to locate the SNPs at 1311 nucleotide position’ should be revised since you did not do it for locating the SNPs.

The authors may be requested to put line number in the revised manuscript.

This manuscript should undergo a significant revision before it could be accepted.

7. PLOS authors have the option to publish the peer review history of their article (what does this mean?). If published, this will include your full peer review and any attached files.

Reviewer #2: **Yes: **Reza Talebi

Reviewer #3: No

---

## [Editor Report · Decision Letter 2]

8 Nov 2022

PONE-D-22-16054R2Designed and Validated Novel Allele-Specific Primer to Differentiate Kernel Row Number (KRN) in Tropical Field Corn

PLOS ONE

Dear Dr. Mukri,

Thank you for submitting your manuscript to PLOS ONE. After careful consideration, we feel that it has merit but does not fully meet PLOS ONE’s publication criteria as it currently stands. Therefore, we invite you to submit a revised version of the manuscript that addresses the points raised during the review process.

I went through your revised manuscript and your response to the comments of the reviewer 3. 

I noticed several problems in this revised manuscripts which should be addressed before it can be accepted. These are listed below.1.
 You mentioned in response to Query no. 1 that “After a thorough review, 13 genes along with fea2 were selected for one particular project. Among the sequence variation, fea2.1 showed SNP that may possibly differentiate high and Low KRN in maize”. I don’t see any where (Materials and Methods or Results), you discussed about it. If that is the case, you should show those genes and sequences to support your statement.2.
I agree with the reviewer 3 that amplicon sequencing is not a next generation sequencing (NGS) technology. Next-generation sequencing (NGS) is a massively parallel sequencing technology that offers ultra-high throughput, scalability, and speed. Some references are: (a) Goodwin et al. 2016. Coming of age: ten years of next generation sequencing technologies. Nature Review Genetics 2016, Volume 17, pp 333-351; (b) Slatko et al. 2018. Overview of Next Generation Sequencing Technologies. Current Protocols in Mol Biol 2018 Vol 122(1):e59.In Line no. 261, you mentioned that you did amplicon sequencing using the primers designed for fea2 gene. However in line # 273, you mentioned about adapter trimming. Where from the adapter come into picture if you did not make any libraries for sequencing. For example. Whole genome sequencing. BWA (https://bio-bwa.sourceforge.net/) is used commonly for whole genome alignment with a large reference genome.In the current write up, it is not yet clear what exactly was done because this methodology does not make sense. I would suggest not to use terminology like ‘Next generation Sequencing’.3.
Line #105: You targeted coding sequences of the gene. Could you please discuss about the gene. For ex. Length, no. of exons, and other molecular characteristics. Is 1842 bp amplicon from a single exon?4.
Supplementary files and Tables were not cited in the Text. It was hard to track these information because these tables and figures were not placed in sequence. I will suggest to put all supplementary table in one file and all suppl figures in another file with suppl table or suppl. Figure 1 first and so on.5.
In supplementary Table 4, please expand the abbreviations in the footnotes. IPCA1, IPCA2, ASVi, ASVi rank etc.6.
Query no. 5: Please mention which suppl table contains information about 197 diverse inbred lines.7.
Since the reviewers will first look at your responses before the manuscript, it is advisable to give a detailed response regarding the changes made in the manuscript. Please submit your revised manuscript by Dec 23 2022 11:59PM. If you will need more time than this to complete your revisions, please reply to this message or contact the journal office at plosone@plos.org. Please include the following items when submitting your revised manuscript:A rebuttal letter that responds to each point raised by the academic editor and reviewer(s). You should upload this letter as a separate file labeled 'Response to Reviewers'.A marked-up copy of your manuscript that highlights changes made to the original version. You should upload this as a separate file labeled 'Revised Manuscript with Track Changes'.An unmarked version of your revised paper without tracked changes. You should upload this as a separate file labeled 'Manuscript'.

We look forward to receiving your revised manuscript.

Kind regards,

Prasanta K. Subudhi, Ph.D.

Academic Editor

PLOS ONE

Additional Editor Comments (if provided):

Major revision
---

## [Editor Report · Decision Letter 3]

20 Dec 2022

PONE-D-22-16054R3Designed and Validated Novel Allele-Specific Primer to Differentiate Kernel Row Number (KRN) in Tropical Field CornPLOS ONE

Dear Dr. Mukri,

Thank you for submitting your manuscript to PLOS ONE. After careful consideration, we feel that it has merit but does not fully meet PLOS ONE’s publication criteria as it currently stands. Therefore, we invite you to submit a revised version of the manuscript that addresses the points raised during the review process.

Although this manuscript has been revised several times, the authors are nor heeding to the suggestions of the reviewers or the editor. We want to give you one more opportunity to address the following concerns. The manuscript should be thoroughly checked for grammatical errors besides the following concerns which should be addressed. •
Line 18: the reference to next generation sequencing should be removed and sentence revised.•
Line 21-22: There is no subject in this sentence. Revise the whole Methodology and principal finding section. This is the most important part of the abstract.•
Line27-28: Delete ‘The utilization of these natural variations in breeding programs is expected to augment the genetic gain.’•
Line 29: Delete ‘easy’•
Line 32-33: Delete ‘as the KRN …. Fairly simple’.•
Line 55-56: Delete ‘Studying the genetic control of some traits of maize ear using molecular markers,”•
Line 99: Similar, finding in accord with Chetan et al., 2021[196]. This is not grammatically correct.•
Line 105: Why ‘anticipated’? are you not sure?•
Remove s from ‘understands’•
Line 374: Table 4 should be deleted since these information can be written in text.•
Figures are numbered 1, 3, 5, 7, 8, 9: It seems the numbers were not corrected in revised manuscript. The correct Figure numbers should be cited in the text.•
At the end (before Acknowledgement section), you listed only 2 supplementary tables. But 6 supplementary tables were cited in text. Please provide the list of Supplementary Tables and their captions.

We look forward to receiving your revised manuscript.

Kind regards,

Prasanta K. Subudhi, Ph.D.

Academic Editor

PLOS ONE

Journal Requirements:

Additional Editor Comments (if provided):

Minor revision

---

## [Author Response · Author response to Decision Letter 3]

21 Jan 2023

Response has been made and letter attached.

---

## [Editor Report · Decision Letter 4]

26 Jan 2023

PONE-D-22-16054R4Designed and Validated Novel Allele-Specific Primer to Differentiate Kernel Row Number (KRN) in Tropical Field CornPLOS ONE

Dear Dr. Mukri,

Thank you for submitting your manuscript to PLOS ONE. After careful consideration, we feel that it has merit but does not fully meet PLOS ONE’s publication criteria as it currently stands. Therefore, we invite you to submit a revised version of the manuscript that addresses the points raised during the review process. 

 Following revisions are requested.1. Line18: Sequencing should start with a small letter.2. Line 21: Delete ".".3. Line 33: insert after for 'improving'4.  After rows, add 'and most of these QTLs are with additive effect'.5. Line 93 and 94: Keep only one decimal6. Line 105: Delete 'is'.7. Line 126: End the sentence after AI536 and start with 'However'8. Line 202 and 203: The sentence lack a verb. revise.9. Line 274: revise to 'amplicons were"10. Line 338: Change 'both in ' to 'in both'11. Line 340: Replace 'the experimentation' with 'this study'.12. Line 379: Delete Table 4 and from the text. (Line 240) and mention in the text .13. In revised supplementary tables: It is mentioned as Supplementary 5 and 6. Revise table caption accordingly.14. Make sure that all tables, figures, and supplementary tables and figures are cited in the text. Please submit your revised manuscript by February 3, 2023. If you will need more time than this to complete your revisions, please reply to this message or contact the journal office at plosone@plos.org. Please include the following items when submitting your revised manuscript:A rebuttal letter that responds to each point raised by the academic editor and reviewer(s). You should upload this letter as a separate file labeled 'Response to Reviewers'.A marked-up copy of your manuscript that highlights changes made to the original version. You should upload this as a separate file labeled 'Revised Manuscript with Track Changes'.An unmarked version of your revised paper without tracked changes. You should upload this as a separate file labeled 'Manuscript'.If applicable, we recommend that you deposit your laboratory protocols in protocols.io to enhance the reproducibility of your results. Protocols.io assigns your protocol its own identifier (DOI) so that it can be cited independently in the future. For instructions see: https://journals.plos.org/plosone/s/submission-guidelines#loc-laboratory-protocols. Additionally, PLOS ONE offers an option for publishing peer-reviewed Lab Protocol articles, which describe protocols hosted on protocols.io. Read more information on sharing protocols at https://plos.org/protocols?utm_medium=editorial-email&utm_source=authorletters&utm_campaign=protocols.

We look forward to receiving your revised manuscript.

Kind regards,

Prasanta K. Subudhi, Ph.D.

Academic Editor

PLOS ONE

Journal Requirements:

Minor revision
---

## [Author Response · Author response to Decision Letter 4]

21 Mar 2023

The corrections suggested on 26 Jan. is addressed accordingly.

---

## [Editor Report · Decision Letter 5]

28 Mar 2023

Designed and Validated Novel Allele-Specific Primer to Differentiate Kernel Row Number (KRN) in Tropical Field Corn

PONE-D-22-16054R5

Dear Dr. Mukri,

We’re pleased to inform you that your manuscript has been judged scientifically suitable for publication and will be formally accepted for publication once it meets all outstanding technical requirements.

Kind regards,

Prasanta K. Subudhi, Ph.D.

Academic Editor

PLOS ONE

Additional Editor Comments (optional):

Accept
---

## [Editor Report · Acceptance letter]

3 Apr 2023

PONE-D-22-16054R5 

Designed and Validated Novel Allele-Specific Primer to Differentiate Kernel Row Number (KRN) in Tropical Field Corn 

Dear Dr. Mukri:

I'm pleased to inform you that your manuscript has been deemed suitable for publication in PLOS ONE. Congratulations! Your manuscript is now with our production department. 

Kind regards, 

on behalf of

Dr. Prasanta K. Subudhi 

Academic Editor

PLOS ONE